# To Trust Or Not To Trust A Classifier

**Heinrich Jiang**[*]
Google Research
heinrichj@google.com

**Been Kim**
Google Brain
beenkim@google.com

**Melody Y. Guan**[†]
Stanford University
mguan@stanford.edu

**Maya Gupta**
Google Research
mayagupta@google.com

## Abstract

Knowing when a classifier's prediction can be trusted is useful in many applications and critical for safely using AI. While the bulk of the effort in machine learning research has been towards improving classifier performance, understanding when a classifier's predictions should and should not be trusted has received far less attention. The standard approach is to use the classifier's discriminant or confidence score; however, we show there exists an alternative that is more effective in many situations. We propose a new score, called the *trust score*, which measures the agreement between the classifier and a modified nearest-neighbor classifier on the testing example. We show empirically that high (low) trust scores produce surprisingly high precision at identifying correctly (incorrectly) classified examples, consistently outperforming the classifier's confidence score as well as many other baselines. Further, under some mild distributional assumptions, we show that if the trust score for an example is high (low), the classifier will likely agree (disagree) with the Bayes-optimal classifier. Our guarantees consist of non-asymptotic rates of statistical consistency under various nonparametric settings and build on recent developments in topological data analysis.

## 1 Introduction

Machine learning (ML) is a powerful and widely-used tool for making potentially important decisions, from product recommendations to medical diagnosis. However, despite ML's impressive performance, it makes mistakes, with some more costly than others. As such, ML trust and safety is an important theme [1, 2, 3]. While improving overall accuracy is an important goal that the bulk of the effort in ML community has been focused on, it may not be enough: we need to also better understand the strengths and limitations of ML techniques.

This work focuses on one such challenge: knowing whether a classifier's prediction for a test example can be trusted or not. Such trust scores have practical applications. They can be directly shown to users to help them gauge whether they should trust the AI system. This is crucial when a model's prediction influences important decisions such as a medical diagnosis, but can also be helpful even in low-stakes scenarios such as movie recommendations. Trust scores can be used to override the classifier and send the decision to a human operator, or to prioritize decisions that human operators should be making. Trust scores are also useful for monitoring classifiers to detect distribution shifts that may mean the classifier is no longer as useful as it was when deployed.

---

[*]All authors contributed equally.

[†]Work done while intern at Google Research.

An open-source implementation of Trust Scores can be found here: https://github.com/google/TrustScore

A standard approach to deciding whether to trust a classifier's decision is to use the classifiers' own reported confidence or score, e.g. probabilities from the softmax layer of a neural network, distance to the separating hyperplane in support vector classification, mean class probabilities for the trees in a random forest. While using a model's own implied confidences appears reasonable, it has been shown that the raw confidence values from a classifier are poorly calibrated [4, 5]. Worse yet, even if the scores are calibrated, the ranking of the scores itself may not be reliable. In other words, a higher confidence score from the model does not necessarily imply higher probability that the classifier is correct, as shown in [6, 7, 8]. A classifier may simply not be the best judge of its own trustworthiness.

In this paper, we use a set of labeled examples (e.g. training data or validation data) to help determine a classifier's trustworthiness for a particular testing example. First, we propose a simple procedure that reduces the training data to a high density set for each class. Then we define the *trust score*—the ratio between the distance from the testing sample to the nearest class different from the predicted class and the distance to the predicted class—to determine whether to trust that classifier prediction.

Theoretically, we show that high/low trust scores correspond to high probability of agreement/disagreement with the Bayes-optimal classifier. We show finite-sample estimation rates when the data is full-dimension and supported on or near a low-dimensional manifold. Interestingly, we attain bounds that depend only on the lower manifold dimension and independent of the ambient dimension without any changes to the procedure or knowledge of the manifold. To our knowledge, these results are new and may be of independent interest.

Experimentally, we found that the trust score better identifies correctly-classified points for low and medium-dimension feature spaces than the model itself. However, high-dimensional feature spaces were more challenging, and we demonstrate that the trust score's utility depends on the vector space used to compute the trust score differences.

## 2   Related Work

One related line of work is that of confidence calibration, which transforms classifier outputs into values that can be interpreted as probabilities, e.g. [9, 10, 11, 4]. In recent work, [5] explore the structured prediction setting, and [12] obtain confidence estimates by using ensembles of networks. These calibration techniques typically only use the the model's reported score (and the softmax layer in the case of a neural network) for calibration, which notably preserves the rankings of the classifier scores. Similarly, [13] considered using the softmax probabilities for the related problem of identifying misclassifications and mislabeled points.

Recent work explored estimating uncertainty for Bayesian neural networks and returning a distribution over the outputs [14, 15]. The proposed trust score does not change the network structure (nor does it assume any structure) and gives a single score, rather than a distribution over outputs as the representation of uncertainty.

The problem of classification with a reject option or learning with abstention [16, 17, 18, 19, 20, 21, 22] is a highly related framework where the classifier is allowed to abstain from making a prediction at a certain cost. Typically such methods jointly learn the classifier and the rejection function. Note that the interplay between classification rate and reject rate is studied in many various forms e.g. [23, 24, 25, 26, 27, 28, 29, 30, 31, 32]. Our paper assumes an already trained and possibly black-box classifier and learns the confidence scores separately, but we do not explicitly learn the appropriate rejection thresholds.

Whether to trust a classifier also arises in the setting where one has access to a sequence of classifiers, but there is some cost to evaluating each classifier, and the goal is to decide after evaluating each classifier in the sequence if one should trust the current classifier decision enough to stop, rather than evaluating more classifiers in the sequence (e.g. [33, 34, 35]). Those confidence decisions are usually based on whether the current classifier score will match the classification of the full sequence.

Experimentally we find that the vector space used to compute the distances in the trust score matters, and that computing trust scores on more-processed layers of a deep model generally works better. This observation is similar to the work of Papernot and McDaniel [36], who use $k$-NN regression on the intermediate representations of the network which they showed enhances robustness to adversarial attacks and leads to better calibrated uncertainty estimations.

Our work builds on recent results in topological data analysis. Our method to filter low-density points estimates a particular density level-set given a parameter $\alpha$, which aims at finding the level-set that contains $1 - \alpha$ fraction of the probability mass. Level-set estimation has a long history [37, 38, 39, 40, 41, 42]. However such works assume knowledge of the density level, which is difficult to determine in practice. We provide rates for Algorithm 1 in estimating the appropriate level-set corresponding to $\alpha$ without knowledge of the level. The proxy $\alpha$ offers a more intuitive parameter compared to the density value, is used for level-set estimation. Our analysis is also done under various settings including when the data lies near a lower dimensional manifold and we provide rates that depend only on the lower dimension.

## 3 Algorithm: The Trust Score

Our approach proceeds in two steps outlined in Algorithm 1 and 2. We first pre-process the training data, as described in Algorithm 1, to find the $\alpha$-*high-density-set* of each class, which is defined as the training samples within that class after filtering out $\alpha$-fraction of the samples with lowest density (which may be outliers):

**Definition 1** ($\alpha$-high-density-set). *Let $0 \leq \alpha < 1$ and $f$ be a continuous density function with compact support $\mathcal{X} \subseteq \mathbb{R}^D$. Then define $H_\alpha(f)$, the $\alpha$-high-density-set of $f$, to be the $\lambda_\alpha$-level set of $f$, defined as $\{x \in \mathcal{X} : f(x) \geq \lambda_\alpha\}$ where $\lambda_\alpha := \inf \left\{ \lambda \geq 0 : \int_{\mathcal{X}} 1 \left[ f(x) \leq \lambda \right] f(x) dx \geq \alpha \right\}$.*

In order to approximate the $\alpha$-high-density-set, Algorithm 1 filters the $\alpha$-fraction of the sample points with lowest *empirical* density, based on $k$-nearest neighbors. This data filtering step is independent of the given classifier $h$.

Then, the second step: given a testing sample, we define its *trust score* to be the ratio between the distance from the testing sample to the $\alpha$-high-density-set of the nearest class different from the predicted class, and the distance from the test sample to the $\alpha$-high-density-set of the class predicted by $h$, as detailed in Algorithm 2. The intuition is that if the classifier $h$ predicts a label that is considerably farther than the closest label, then this is a warning that the classifier may be making a mistake.

Our procedure can thus be viewed as a comparison to a modified nearest-neighbor classifier, where the modification lies in the initial filtering of points not in the $\alpha$-high-density-set for each class.

**Remark 1.** *The distances can be computed with respect to any representation of the data. For example, the raw inputs, an unsupervised embedding of the space, or the activations of the intermediate representations of the classifier. Moreover, the nearest-neighbor distance can be replaced by other distance measures, such as $k$-nearest neighbors or distance to a centroid.*

---

**Algorithm 1** Estimating $\alpha$-high-density-set

Parameters: $\alpha$ (density threshold), $k$.
Inputs: Sample points $X := \{x_1, .., x_n\}$ drawn from $f$.
Define $k$-NN radius $r_k(x) := \inf\{r > 0 : |B(x,r) \cap X| \geq k\}$ and let $\varepsilon := \inf\{r > 0 : |\{x \in X : r_k(x) > r\}| \leq \alpha \cdot n\}$.
**return** $\widehat{H_\alpha}(f) := \{x \in X : r_k(x) \leq \varepsilon\}$.

---

**Algorithm 2** Trust Score

Parameters: $\alpha$ (density threshold), $k$.
Input: Classifier $h : \mathcal{X} \to \mathcal{Y}$. Training data $(x_1, y_1), ..., (x_n, y_n)$. Test example $x$.
For each $\ell \in \mathcal{Y}$, let $\widehat{H_\alpha}(f_\ell)$ be the output of Algorithm 1 with parameters $\alpha, k$ and sample points $\{x_j : 1 \leq j \leq n, y_j = \ell\}$. Then, return the trust score, defined as:

$$\xi(h, x) := d\left(x, \widehat{H_\alpha}(f_{\widetilde{h}(x)})\right) / d\left(x, \widehat{H_\alpha}(f_{h(x)})\right),$$

where $\widetilde{h}(x) = \mathrm{argmin}_{l \in \mathcal{Y}, l \neq h(x)} d\left(x, \widehat{H_\alpha}(f_l)\right)$.

---

The method has two hyperparameters: $k$ (the number of neighbors, such as in $k$-NN) and $\alpha$ (fraction of data to filter) to compute the empirical densities. We show in theory that $k$ can lie in a wide range

and still give us the desired consistency guarantees. Throughout our experiments, we fix $k = 10$, and use cross-validation to select $\alpha$ as it is data-dependent.

**Remark 2.** *We observed that the procedure was not very sensitive to the choice of $k$ and $\alpha$. As will be shown in the experimental section, for efficiency on larger datasets, we skipped the initial filtering step of Algorithm 1 (leading to a hyperparameter-free procedure) and obtained reasonable results. This initial filtering step can also be replaced by other strategies. One such example is filtering examples whose labels have high disagreement amongst its neighbors, which is implemented in the open-source code release but not experimented with here.*

## 4 Theoretical Analysis

In this section, we provide theoretical guarantees for Algorithms 1 and 2. Due to space constraints, all the proofs are deferred to the Appendix. To simplify the main text, we state our results treating $\delta$, the confidence level, as a constant. The dependence on $\delta$ in the rates is made explicit in the Appendix.

We show that Algorithm 1 is a statistically consistent estimator of the $\alpha$-high-density-level set with finite-sample estimation rates. We analyze Algorithm 1 in three different settings: when the data lies on (i) a full-dimensional $\mathbb{R}^D$; (ii) an unknown lower dimensional submanifold embedded in $\mathbb{R}^D$; and (iii) an unknown lower dimensional submanifold with full-dimensional noise.

For setting (i), where the data lies in $\mathbb{R}^D$, the estimation rate has a dependence on the dimension $D$, which may be unattractive in high-dimensional situations: this is known as the curse of dimensionality, suffered by density-based procedures in general. However, when the data has low intrinsic dimension in (ii), it turns out that, remarkably, without any changes to the procedure, the estimation rate depends on the lower dimension $d$ and is *independent* of the ambient dimension $D$. However, in realistic situations, the data may not lie *exactly* on a lower-dimensional manifold, but *near* one. This reflects the setting of (iii), where the data essentially lies on a manifold but has general full-dimensional noise so the data is overall full-dimensional. Interestingly, we show that we still obtain estimation rates depending only on the manifold dimension and independent of the ambient dimension; moreover, we do not require knowledge of the manifold nor its dimension to attain these rates.

We then analyze Algorithm 2, and establish the culminating result of Theorem 4: for labeled data distributions with well-behaved class margins, when the trust score is large, the classifier likely agrees with the Bayes-optimal classifier, and when the trust score is small, the classifier likely disagrees with the Bayes-optimal classifier. If it turns out that even the Bayes-optimal classifier has high-error in a certain region, then any classifier will have difficulties in that region. Thus, Theorem 4 does not guarantee that the trust score can predict misclassification, but rather that it can predict when the classifier is making an unreasonable decision.

### 4.1 Analysis of Algorithm 1

We require the following regularity assumptions on the boundaries of $H_\alpha(f)$, which are standard in analyses of level-set estimation [40]. Assumption 1.1 ensures that the density around $H_\alpha(f)$ has both smoothness and curvature. The upper bound gives smoothness, which is important to ensure that our density estimators are accurate for our analysis (we only require this smoothness near the boundaries and not globally). The lower bound ensures curvature: this ensures that $H_\alpha(f)$ is salient enough to be estimated. Assumption 1.2 ensures that $H_\alpha(f)$ does not get arbitrarily thin anywhere.

**Assumption 1** ($\alpha$-high-density-set regularity). *Let $\beta > 0$. There exists $\check{C}_\beta, \hat{C}_\beta, \beta, r_c, r_0, \rho > 0$ s.t.*

1. *$\check{C}_\beta \cdot d(x, H_\alpha(f))^\beta \leq |\lambda_\alpha - f(x)| \leq \hat{C}_\beta \cdot d(x, H_\alpha(f))^\beta$ for all $x \in \partial H_\alpha(f) + B(0, r_c)$.*

2. *For all $0 < r < r_0$ and $x \in H_\alpha(f)$, we have $Vol(B(x, r)) \geq \rho \cdot r^D$.*

*where $\partial A$ denotes the boundary of a set $A$, $d(x, A) := \inf_{x' \in A} ||x - x'||$, $B(x, r) := \{x' : |x - x'| \leq r\}$ and $A + B(0, r) := \{x : d(x, A) \leq r\}$.*

Our statistical guarantees are under the Hausdorff metric, which ensures a *uniform* guarantee over our estimator: it is a stronger notion of consistency than other common metrics [41, 43].

**Definition 2** (Hausdorff distance). *$d_H(A, B) := \max\{\sup_{x \in A} d(x, B), \sup_{x \in B} d(x, A)\}$.*

We now give the following result for Algorithm 1. It says that as long as our density function satisfies the regularity assumptions stated earlier, and the parameter $k$ lies within a certain range, then we can bound the Hausdorff distance between what Algorithm 1 recovers and $H_\alpha(f)$, the true $\alpha$-high-density set, from an i.i.d. sample drawn from $f$ of size $n$. Then, as $n$ goes to $\infty$, and $k$ grows as a function of $n$, the quantity goes to $0$.

**Theorem 1** (Algorithm 1 guarantees). *Let $0 < \delta < 1$ and suppose that $f$ is continuous and has compact support $\mathcal{X} \subseteq \mathbb{R}^D$ and satisfies Assumption 1. There exists constants $C_l, C_u, C > 0$ depending on $f$ and $\delta$ such that the following holds with probability at least $1 - \delta$. Suppose that $k$ satisfies $C_l \cdot \log n \le k \le C_u \cdot (\log n)^{D(2\beta+D)} \cdot n^{2\beta/(2\beta+D)}$. Then we have*

$$d_H(H_\alpha(f), \widehat{H_\alpha}(f)) \le C \cdot \left( n^{-1/2D} + \log(n)^{1/2\beta} \cdot k^{-1/2\beta} \right).$$

**Remark 3.** *The condition on $k$ can be simplified by ignoring log factors: $\log n \lesssim k \lesssim n^{2\beta/(2\beta+D)}$, which is a wide range. Setting $k$ to its allowed upper bound, we obtain our consistency guarantee of*

$$d_H(H_\alpha(f), \widehat{H_\alpha}(f)) \lesssim \max\{n^{-1/2D}, n^{-1/(2\beta+D)}\}.$$

*The first term is due to the error from estimating the appropriate level given $\alpha$ (i.e. identifying the level $\lambda_\alpha$) and the second term corresponds to the error for recovering the level set given knowledge of the level. The latter term matches the lower bound for level-set estimation up to log factors [39].*

## 4.2 Analysis of Algorithm 1 on Manifolds

One of the disadvantages of Theorem 1 is that the estimation errors have a dependence on $D$, the dimension of the data, which may be highly undesirable in high-dimensional settings. We next improve these rates when the data has a lower intrinsic dimension. Interestingly, we are able to show rates that depend only on the intrinsic dimension of the data, without explicit knowledge of that dimension nor any changes to the procedure. As common to related work in the manifold setting, we make the following regularity assumptions which are standard among works in manifold learning (e.g. [44, 45, 46]).

**Assumption 2** (Manifold Regularity). *$M$ is a $d$-dimensional smooth compact Riemannian manifold without boundary embedded in compact subset $\mathcal{X} \subseteq \mathbb{R}^D$ with bounded volume. $M$ has finite condition number $1/\tau$, which controls the curvature and prevents self-intersection.*

**Theorem 2** (Manifold analogue of Theorem 1). *Let $0 < \delta < 1$. Suppose that density function $f$ is continuous and supported on $M$ and Assumptions 1 and 2 hold. Suppose also that there exists $\lambda_0 > 0$ such that $f(x) \ge \lambda_0$ for all $x \in M$. Then, there exists constants $C_l, C_u, C > 0$ depending on $f$ and $\delta$ such that the following holds with probability at least $1 - \delta$. Suppose that $k$ satisfies $C_l \cdot \log n \le k \le C_u \cdot (\log n)^{d(2\beta'+d)} \cdot n^{2\beta'/(2\beta'+d)}$. where $\beta' := \max\{1, \beta\}$. Then we have*

$$d_H(H_\alpha(f), \widehat{H_\alpha}(f)) \le C \cdot \left( n^{-1/2d} + \log(n)^{1/2\beta} \cdot k^{-1/2\beta} \right).$$

**Remark 4.** *Setting $k$ to its allowed upper bound, we obtain (ignoring log factors),*

$$d_H(H_\alpha(f), \widehat{H_\alpha}(f)) \lesssim \max\{n^{-1/2d}, n^{-1/(2\max\{1,\beta\}+d)}\}.$$

*The first term can be compared to that of the previous result where $D$ is replaced with $d$. The second term is the error for recovering the level set on manifolds, which matches recent rates [42].*

## 4.3 Analysis of Algorithm 1 on Manifolds with Full Dimensional Noise

In realistic settings, the data may not lie exactly on a low-dimensional manifold, but *near* one. We next present a result where the data is distributed along a manifold with additional full-dimensional noise. We make mild assumptions on the noise distribution. Thus, in this situation, the data has intrinsic dimension equal to the ambient dimension. Interestingly, we are still able to show that the rates only depend on the dimension of the manifold and not the dimension of the entire data.

**Theorem 3.** *Let $0 < \eta < \alpha < 1$ and $0 < \delta < 1$. Suppose that distribution $\mathcal{F}$ is a weighted mixture $(1-\eta) \cdot \mathcal{F}_M + \eta \cdot \mathcal{F}_E$ where $\mathcal{F}_M$ is a distribution with continous density $f_M$ supported on a $d$-dimensional manifold $M$ satisfying Assumption 2 and $\mathcal{F}_E$ is a (noise) distribution with continuous density $f_E$ with compact support over $\mathbb{R}^D$ with $d < D$. Suppose also that there exists $\lambda_0 > 0$ such*

that $f_M(x) \geq \lambda_0$ for all $x \in M$ and $H_{\widetilde{\alpha}}(f_M)$ (where $\widetilde{\alpha} := \frac{\alpha - \eta}{1 - \eta}$) satisfies Assumption 1 for density $f_M$. Let $\widehat{H}_\alpha$ be the output of Algorithm 1 on a sample $X$ of size $n$ drawn i.i.d. from $\mathcal{F}$. Then, there exists constants $C_l, C_u, C > 0$ depending on $f_M$, $f_E$, $\eta$, $M$ and $\delta$ such that the following holds with probability at least $1 - \delta$. Suppose that $k$ satisfies $C_l \cdot \log n \leq k \leq C_u \cdot (\log n)^{d(2\beta' + d)} \cdot n^{2\beta'/(2\beta' + d)}$, where $\beta' := \max\{1, \beta\}$. Then we have

$$d_H(H_{\widetilde{\alpha}}(f_M), \widehat{H}_\alpha) \leq C \cdot \left( n^{-1/2d} + \log(n)^{1/2\beta} \cdot k^{-1/2\beta} \right).$$

The above result is compelling because it shows why our methods can work, even in high-dimensions, despite the curse of dimensionality of non-parametric methods. In typical real-world data, even if the data lies in a high-dimensional space, there may be far fewer degrees of freedom. Thus, our theoretical results suggest that when this is true, then our methods will enjoy far better convergence rates – even when the data overall has full intrinsic dimension due to factors such as noise.

## 4.4 Analysis of Algorithm 2: the Trust Score

We now provide a guarantee about the trust score, making the same assumptions as in Theorem 3 for each of the label distributions. We additionally assume that the class distributions are well-behaved in the following sense: that high-density-regions for each of the classes satisfy the property that for any point $x \in \mathcal{X}$, if the ratio of the distance to one class's high-density-region to that of another is smaller by some margin $\gamma$, then it is more likely that $x$'s label corresponds to the former class.

**Theorem 4.** *Let $0 < \eta < \alpha < 1$. Let us have labeled data $(x_1, y_1), ..., (x_n, y_n)$ drawn from distribution $\mathcal{D}$, which is a joint distribution over $\mathcal{X} \times \mathcal{Y}$ where $\mathcal{Y}$ are the labels, $|\mathcal{Y}| < \infty$, and $\mathcal{X} \subseteq \mathbb{R}^D$ is compact. Suppose for each $\ell \in \mathcal{Y}$, the conditional distribution for label $\ell$ satisfies the conditions of Theorem 3 for some manifold and noise level $\eta$. Let $f_{M,\ell}$ be the density of the portion of the conditional distribution for label $\ell$ supported on $M$. Define $M_\ell := H_{\widetilde{\alpha}}(f_\ell)$, where $\widetilde{\alpha} := \frac{\alpha - \eta}{1 - \eta}$ and let $\epsilon_n$ be the maximum Hausdorff error from estimating $M_\ell$ over each $\ell \in \mathcal{Y}$ in Theorem 3. Assume that $\min_{\ell \in \mathcal{Y}} \mathbb{P}_\mathcal{D}(y = \ell) > 0$ to ensure we have samples from each label.*

*Suppose also that for each $x \in \mathcal{X}$, if $d(x, M_i)/d(x, M_j) < 1 - \gamma$ then $\mathbb{P}(y = i|x) > \mathbb{P}(y = j|x)$ for $i, j \in \mathcal{Y}$. That is, if we are closer to $M_i$ than $M_j$ by a ratio of less than $1 - \gamma$, then the point is more likely to be from class $i$. Let $h^*$ be the Bayes-optimal classifier, defined by $h^*(x) := \text{argmax}_{\ell \in \mathcal{Y}} \mathbb{P}(y = \ell|x)$. Then the trust score $\xi$ of Algorithm 2 satisfies the following with high probability uniformly over all $x \in \mathcal{X}$ and all classifiers $h : \mathcal{X} \to \mathcal{Y}$ simultaneously for $n$ sufficiently large depending on $\mathcal{D}$:*

$$\xi(h, x) < 1 - \gamma - \frac{\epsilon_n}{d(x, M_{h(x)}) + \epsilon_n} \cdot \left( \frac{d(x, M_{\widetilde{h}(x)})}{d(x, M_{h(x)})} + 1 \right) \quad \Rightarrow \quad h(x) \neq h^*(x),$$

$$\frac{1}{\xi(h, x)} < 1 - \gamma - \frac{\epsilon_n}{d(x, M_{\widetilde{h}(x)}) + \epsilon_n} \cdot \left( \frac{d(x, M_{h(x)})}{d(x, M_{\widetilde{h}(x)})} + 1 \right) \quad \Rightarrow \quad h(x) = h^*(x).$$

## 5 Experiments

In this section, we empirically test whether trust scores can both detect examples that are incorrectly classified with high precision and be used as a signal to determine which examples are likely correctly classified. We perform this evaluation across (i) different datasets (Sections 5.1 and 5.3), (ii) different families of classifiers (neural network, random forest and logistic regression) (Section 5.1), (iii) classifiers with varying accuracy on the same task (Section 5.2) and (iv) different representations of the data e.g. input data or activations of various intermediate layers in neural network (Section 5.3).

First, we test if testing examples with high trust score corresponds to examples in which the model is correct ("identifying trustworthy examples"). Each method produces a numeric score for each testing example. For each method, we bin the data points by percentile value of the score (i.e. 100 bins). Given a recall percentile level (i.e. the $x$-axis on our plots), we take the performance of the classifier on the bins above the percentile level as the precision (i.e. the $y$-axis). Then, we take the negative of each signal and test if low trust score corresponds to the model being wrong ("identifying suspicious

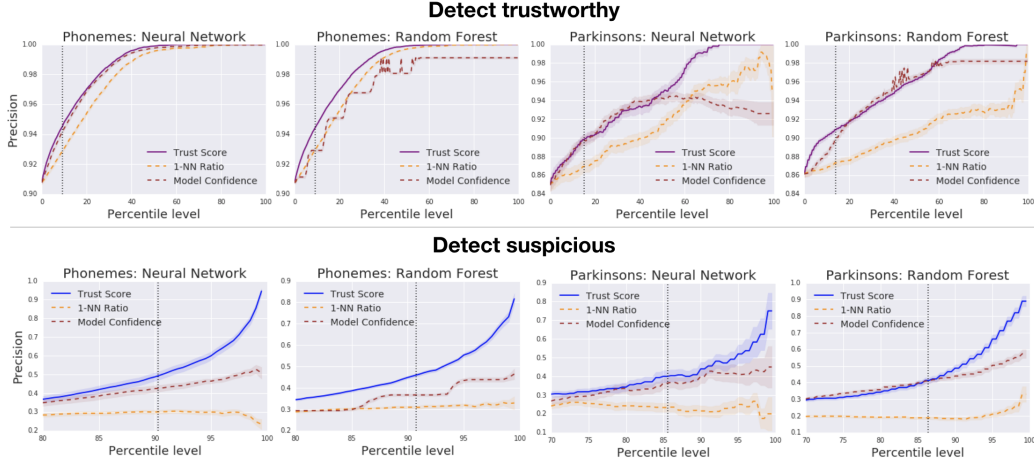

Figure 1: Two example datasets and models. For predicting correctness (top row) the vertical dotted black line indicates error level of the trained classifier. For predicting incorrectness (bottom) the vertical black dotted line is the accuracy rate of the classifier. For detecting trustworthy, for each percentile level, we take the test examples whose trust score was above that percentile level and plot the percentage of those test points that were correctly classified by the classifier, and do the same model confidence and 1-nn ratio. For detecting suspicious, we take the negative of each signal and plot the precision of identifying incorrectly classified examples. Shown are average of 20 runs with shaded standard error band. The trust score consistently attains a higher precision for each given percentile of classifier decision-rejection. Furthermore, the trust score generally shows increasing precision as the percentile level increases, but surprisingly, many of the comparison baselines do not. See the Appendix for the full results.

examples"). Here the $y$-axis is the misclassification rate and the $x$-axis corresponds to decreasing trust score or model confidence.

In both cases, the higher the precision vs percentile curve, the better the method. The vertical black dotted lines in the plots represent the omniscient ideal. For identifying trustworthy examples it is the error rate of the classifier and for identifying suspicious examples" it is the accuracy rate.

The baseline we use in Section is the model's own confidence score, which is similar to the approach of [13]. While calibrating the classifiers' confidence scores (i.e. transforming them into probability estimates of correctness) is an important related work [4, 9], such techniques typically do not change the rankings of the score, at least in the binary case. Since we evaluate the trust score on its precision at a given recall *percentile level*, we are interested in the relative *ranking* of the scores rather than their absolute values. Thus, we do not compare against calibration techniques. There are surprisingly few methods aimed at identifying correctly or incorrectly classified examples with precision at a recall percentile level as noted in [13].

**Choosing Hyperparameters**: The two hyperparameters for the trust score are $\alpha$ and $k$. Throughout the experiments, we fix $k = 10$ and choose $\alpha$ using cross-validation over (negative) powers of 2 on the training set. The metric for cross-validation was optimal performance on detecting suspicious examples at the percentile corresponding to the classifier's accuracy. The bulk of the computational cost for the trust-score is in $k$-nearest neighbor computations for training and 1-nearest neighbor searches for evaluation. To speed things up for the larger datasets MNIST, SVHN, CIFAR-10 and CIFAR-100, we skipped the initial filtering step of Algorithm 1 altogether and reduced the intermediate layers down to 20 dimensions using PCA before being processed by the trust score which showed similar performance. We note that any approximation method (such as approximate instead of exact nearest neighbors) could have been used instead.

## 5.1 Performance on Benchmark UCI Datasets

In this section, we show performance on five benchmark UCI datasets [47], each for three kinds of classifiers (neural network, random forest and logistic regression). Due to space, we only show

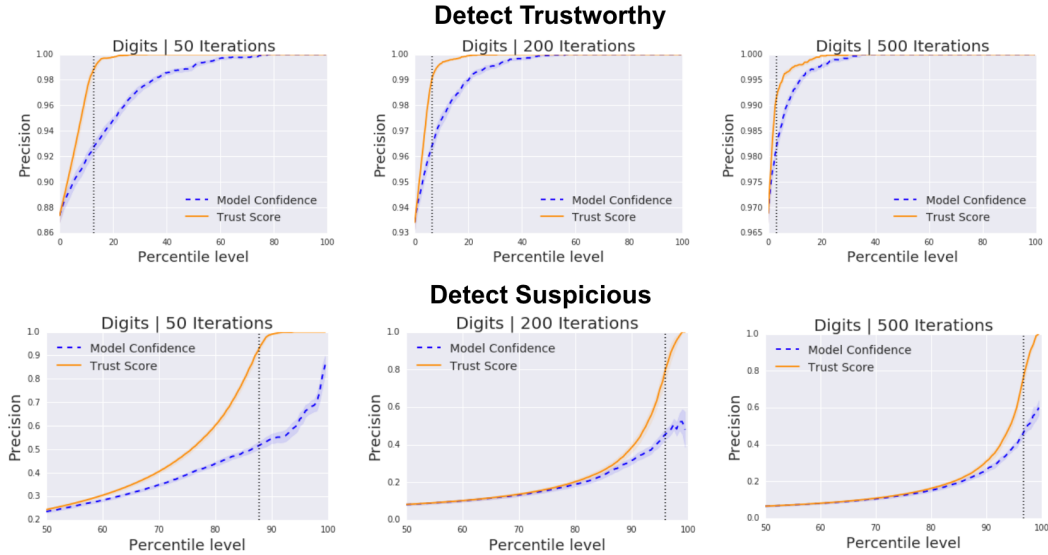

Figure 2: We show the performance of trust score on the Digits dataset for a neural network as we increase the accuracy. As we go from left to right, we train the network with more iterations (each with batch size 50) thus increasing the accuracy indicated by the dotted vertical lines. While the trust score still performs better than model confidence, the amount of improvement diminishes.

two data sets and two models in Figure 1. The rest can be found in the Appendix. For each method and dataset, we evaluated with multiple runs. For each run we took a random stratified split of the dataset into two halves. One portion was used for training the trust score and the other was used for evaluation and the standard error is shown in addition to the average precision across the runs at each percentile level. The results show that our method consistently has a higher precision vs percentile curve than the rest of the methods across the datasets and models. This suggests the trust score considerably improves upon known methods as a signal for identifying trustworthy and suspicious testing examples for low-dimensional data.

In addition to the model's own confidence score, we try one additional baseline, which we call the *nearest neighbor ratio (1-nn ratio)*. It is the ratio between the 1-nearest neighbor distance to the closest and second closest class, which can be viewed as an analogue to the trust score without knowledge of the classifier's hard prediction.

## 5.2 Performance as Model Accuracy Varies

In Figure 2, we show how the performance of trust score changes as the accuracy of the classifier changes (averaged over 20 runs for each condition). We observe that as the accuracy of the model increases, while the trust score still performs better than model confidence, the amount of improvement diminishes. This suggests that as the model improves, the information trust score can provide in addition to the model confidence decreases. However, as we show in Section 5.3, the trust score can still have added value even when the classifier is known to be of high performance on some benchmark larger-scale datasets.

## 5.3 Performance on MNIST, SVHN, CIFAR-10 and CIFAR-100 Datasets

The MNIST handwritten digit dataset [48] consists of 60,000 28×28-pixel training images and 10,000 testing images in 10 classes. The SVHN dataset [49] consists of 73,257 32×32-pixel colour training images and 26,032 testing images and also has 10 classes. The CIFAR-10 and CIFAR-100 datasets [50] both consist of 60,000 32×32-pixel colour images, with 50,000 training images and 10,000 test images. The CIFAR-10 and CIFAR-100 datasets are split evenly between 10 classes and 100 classes respectively.

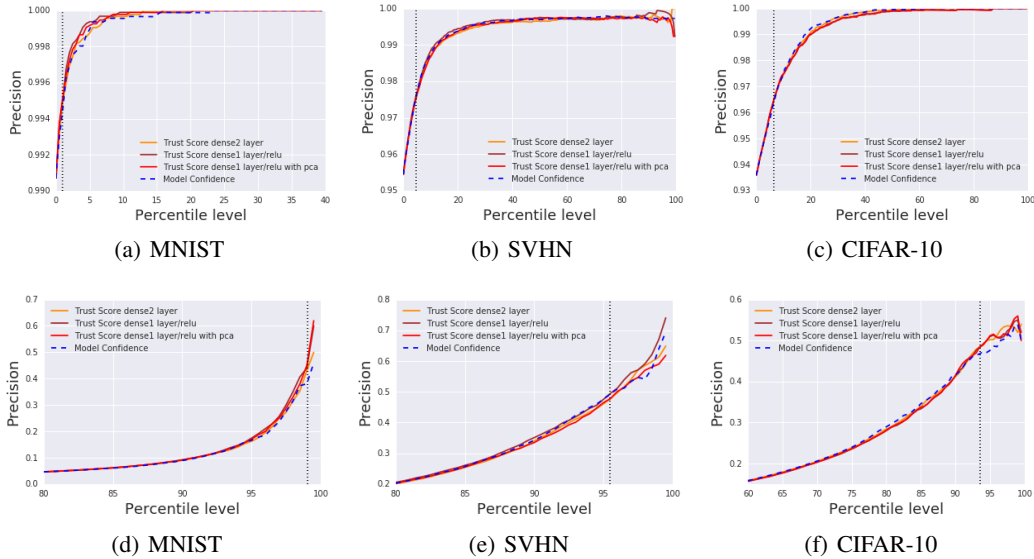

Figure 3: Trust score results using convolutional neural networks on MNIST, SVHN, and CIFAR-10 datasets. Top row is detecting trustworthy; bottom row is detecting suspicious. Full chart with CIFAR-100 (which was essentially a negative result) is shown in the Appendix.

We used a pretrained VGG-16 [51] architecture with adaptation to the CIFAR datasets based on [52]. The CIFAR-10 VGG-16 network achieves a test accuracy of 93.56% while the CIFAR-100 network achieves a test accuracy of 70.48%. We used pretrained, smaller CNNs for MNIST and SVHN. The MNIST network achieves a test accuracy of 99.07% and the SVHN network achieves a test accuracy of 95.45%. All architectures were implemented in Keras [53].

One simple generalization of our method is to use intermediate layers of a neural network as an input instead of the raw $x$. Many prior work suggests that a neural network may learn different representations of $x$ at each layer. As input to the trust score, we tried using 1) the logit layer, 2) the preceding fully connected layer with ReLU activation, 3) this fully connected layer, which has 128 dimensions in the MNIST network and 512 dimensions in the other networks, reduced down to 20 dimensions from applying PCA.

The trust score results on various layers are shown in Figure 3. They suggest that for high dimensional datasets, the trust score may only provide little or no improvement over the model confidence at detecting trustworthy and suspicious examples. All plots were made using $\alpha = 0$; using cross-validation to select a different $\alpha$ did not improve trust score performance. We also did not see much difference from using different layers.

**Conclusion**:

In this paper, we provide the *trust score*: a new, simple, and effective way to judge if one should trust the prediction from a classifier. The trust score provides information about the relative positions of the datapoints, which may be lost in common approaches such as the model confidence when the model is trained using SGD. We show high-probability non-asymptotic statistical guarantees that high (low) trust scores correspond to agreement (disagreement) with the Bayes-optimal classifier under various nonparametric settings, which build on recent results in topological data analysis. Our empirical results across many datasets, classifiers, and representations of the data show that our method consistently outperforms the classifier's own reported confidence in identifying trustworthy and suspicious examples in low to mid dimensional datasets. The theoretical and empirical results suggest that this approach may have important practical implications in low to mid dimension settings.

---

https://github.com/geifmany/cifar-vgg
https://github.com/EN10/KerasMNIST
https://github.com/tohinz/SVHN-Classifier

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
