[Supplementary Material]

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

# Appendix

## A    Supporting results for Theorem 1 Proof

We need the following result giving guarantees on the empirical balls.

**Lemma 1** (Uniform convergence of balls [54]). *Let $\mathcal{F}$ be the distribution corresponding to $f$ and $\mathcal{F}_n$ be the empirical distribution corresponding to the sample $X$. Pick $0 < \delta < 1$. Assume that $k \geq D \log n$. Then with probability at least $1 - \delta$, for every ball $B \subset \mathbb{R}^D$ we have*

$$\mathcal{F}(B) \geq C_{\delta,n} \frac{\sqrt{D \log n}}{n} \Rightarrow \mathcal{F}_n(B) > 0$$

$$\mathcal{F}(B) \geq \frac{k}{n} + C_{\delta,n} \frac{\sqrt{k}}{n} \Rightarrow \mathcal{F}_n(B) \geq \frac{k}{n}$$

$$\mathcal{F}(B) \leq \frac{k}{n} - C_{\delta,n} \frac{\sqrt{k}}{n} \Rightarrow \mathcal{F}_n(B) < \frac{k}{n},$$

*where $C_{\delta,n} = 16 \log(2/\delta)\sqrt{D \log n}$*

**Remark 5.** *For the rest of the paper, many results are qualified to hold with probability at least $1 - \delta$. This is precisely the event in which Lemma 1 holds.*

**Remark 6.** *If $\delta = 1/n$, then $C_{\delta,n} = O((\log n)^{3/2})$.*

To analyze Algorithm 1, we use the $k$-NN density estimator[55], defined below.

**Definition 3.** *Define the k-NN radius of $x \in \mathbb{R}^D$ as*

$$r_k(x) := \inf\{r > 0 : |X \cap B(x, r)| \geq k\},$$

**Definition 4** (k-NN Density Estimator).

$$f_k(x) := \frac{k}{n \cdot v_D \cdot r_k(x)^D},$$

*where $v_D$ is the volume of a unit ball in $\mathbb{R}^D$.*

We will use bounds on the $k$-NN density estimator from [56], which are repeated here.

Define the following one-sided modulus of continuity which characterizes how much the density increases locally:

$$\hat{r}(\epsilon, x) := \sup \left\{ r : \sup_{x' \in B(x,r)} f(x') - f(x) \leq \epsilon \right\}.$$

**Lemma 2** (Lemma 3 of [56]). *Suppose that $k \geq 4C_{\delta,n}^2$. Then with probability at least $1 - \delta$, the following holds for all $x \in \mathbb{R}^D$ and $\epsilon > 0$.*

$$f_k(x) < \left(1 + 2\frac{C_{\delta,n}}{\sqrt{k}}\right)(f(x) + \epsilon),$$

*provided $k$ satisfies $v_D \cdot \hat{r}(x, \epsilon)^D \cdot (f(x) + \epsilon) \geq \frac{k}{n} + C_{\delta,n}\frac{\sqrt{k}}{n}$.*

Analogously, define the following which characterizes how much the density decreases locally:

$$\check{r}(\epsilon, x) := \sup \left\{ r : \sup_{x' \in B(x,r)} f(x) - f(x') \leq \epsilon \right\}.$$

**Lemma 3** (Lemma 4 of [56]). *Suppose that $k \geq 4C_{\delta,n}^2$. Then with probability at least $1 - \delta$, the following holds for all $x \in \mathbb{R}^D$ and $\epsilon > 0$.*

$$f_k(x) \geq \left(1 - 2\frac{C_{\delta,n}}{\sqrt{k}}\right)(f(x) - \epsilon),$$

*provided $k$ satisfies $v_D \cdot \check{r}(x, \epsilon)^D \cdot (f(x) - \epsilon) \geq \frac{k}{n} - C_{\delta,n}\frac{\sqrt{k}}{n}$.*

# B Proof of Theorem 1

In this section, we assume the conditions of Theorem 1. We first show that $\lambda_\alpha$, that is the density level corresponding to the $\alpha$-high-density-set, is smooth in $\alpha$.

**Lemma 4.** *There exists constants $C_1, r_1 > 0$ depending on $f$ such that the following holds for all $0 < \epsilon < r_1$ such that*

$$0 < \lambda_\alpha - \lambda_{\alpha-\epsilon} \leq C_1 \epsilon^{\beta/D} \text{ and } 0 < \lambda_{\alpha+\epsilon} - \lambda_\alpha \leq C_1 \epsilon^{\beta/D}.$$

*Proof.* We have

$$\epsilon = \int_{\mathcal{X}} 1[\lambda_{\alpha-\epsilon} < f(x) \leq \lambda_\alpha] \cdot f(x) dx \geq \lambda_{\alpha-\epsilon} \int_{\mathcal{X}} 1[\lambda_{\alpha-\epsilon} < f(x) \leq \lambda_\alpha] dx,$$

where the first equality holds by definition. Choosing $\epsilon$ sufficiently small such that Assumption 1 holds, we have

$$\lambda_{\alpha-\epsilon} \int_{\mathcal{X}} 1[\lambda_{\alpha-\epsilon} < f(x) \leq \lambda_\alpha] dx$$
$$\geq \lambda_{\alpha-\epsilon} \cdot \text{Vol}\left(\left(H_\alpha(x) + B\left(0, ((\lambda_{\alpha-\epsilon} - \lambda_\alpha)/\widehat{C}_\beta)^{1/\beta}\right)\right) \backslash H_\alpha(f)\right)$$
$$\geq \lambda_{\alpha-\epsilon} \cdot C'((\lambda_{\alpha-\epsilon} - \lambda_\alpha)/\widehat{C}_\beta)^{D/\beta},$$

where the last inequality holds for some constant $C'$ depending on $f$ and Vol is the volume w.r.t. to the Lebesgue measure in $\mathbb{R}^D$. It then follows that

$$\lambda_{\alpha-\epsilon} - \lambda_\alpha \leq \widehat{C}_\beta \left(\frac{\epsilon}{\lambda_{\alpha-\epsilon} \cdot C'}\right)^{\beta/D},$$

and the result for the first part follows by taking $C_1 \leq \widehat{C}_\beta \cdot (\lambda_{\alpha-r_1} \cdot C')^{-\beta/D}$ and $r_1 < \alpha$. Showing that $0 < \lambda_{\alpha+\epsilon} - \lambda_\alpha \leq C_1 \epsilon^{\beta/D}$ can be done analogously and is omitted here. $\square$

The next result gets a handle on the density level corresponding to $\alpha$ returned by Algorithm 1.

**Lemma 5.** *Let $0 < \delta < 1$. Let $\widehat{\varepsilon}$ be the $\varepsilon$ setting chosen by Algorithm 1. Define*

$$\widehat{\lambda_\alpha} := \frac{k}{v_D \cdot n \cdot \widehat{\varepsilon}^D}.$$

*Then, with probability at least $1 - \delta$, we have there exist constant $C_1 > 0$ depending on $f$ such that for $n$ sufficiently large depending on $f$, we have*

$$|\widehat{\lambda_\alpha} - \lambda_\alpha| \leq C_1 \left(\left(\sqrt{\frac{\log(1/\delta)}{n}}\right)^{\beta/D} + \frac{\log(1/\delta)\sqrt{\log n}}{\sqrt{k}}\right).$$

*Proof.* Let $\tilde{\alpha} > 0$. Then, we have that if $x \sim f$, then $\mathbb{P}(x \in H_{\tilde{\alpha}}(f)) = 1 - \tilde{\alpha}$. Thus, the probability that a sample point falls in $H_{\tilde{\alpha}}(f)$ is a Bernoulli random variable with probability $1 - \tilde{\alpha}$. Hence, by Hoeffding's inequality, we have that there exist constant $C' > 0$ such that

$$\mathbb{P}\left(1 - \tilde{\alpha} - C'\sqrt{\frac{\log(1/\delta)}{n}} \leq \frac{|H_{\tilde{\alpha}}(f) \cap X|}{n} \leq 1 - \tilde{\alpha} + C'\sqrt{\frac{\log(1/\delta)}{n}}\right) \geq 1 - \delta/4.$$

Then it follows that choosing $\alpha_U := \alpha + C'\sqrt{\frac{\log(1/\delta)}{n}}$ we get

$$\mathbb{P}\left(\frac{|H_{\alpha_U}(f) \cap X|}{n} \leq 1 - \alpha\right) \geq 1 - \delta/4.$$

Similarly, choosing $\alpha_L = \alpha - C'\sqrt{\frac{\log(1/\delta)}{n}}$ gives us

$$\mathbb{P}\left(\frac{|H_{\alpha_L}(f) \cap X|}{n} \geq 1 - \alpha\right) \geq 1 - \delta/4.$$

Next, define
$$H_\alpha^{upper}(f) := \{x \in X : f_k(x) \geq \lambda_\alpha - \epsilon\},$$
where $\epsilon > 0$ will be chosen later in order for $\widehat{H}_\alpha(f) \subseteq H_\alpha^{upper}(f)$. By Lemma 4, there exists $C_2, r_1 > 0$ depending on $f$ such that for $\widehat{\varepsilon} < r_1$ (which holds for $n$ sufficiently large depending on $f$ by Lemma 1), we have $\lambda_\alpha - C_2 \left( \sqrt{\frac{\log(1/\delta)}{n}} \right)^{\beta/D} \leq \lambda_{\alpha_L}$. As such, it suffices to choose $\epsilon$ such that for all $x \in \mathcal{X}$ such that if $f(x) \geq \lambda_\alpha - C_2 \left( \sqrt{\frac{\log(1/\delta)}{n}} \right)^{\beta/D}$ then $f_k(x) \geq \lambda_\alpha - \epsilon$. This is because $\{x \in X : f_k(x) \geq \lambda_\alpha - \epsilon\}$ would contain $H_{\alpha_L}(f) \cap X$, which we showed earlier contains at least $1 - \alpha$ fraction of the samples. Define $\epsilon_0$ such that $\epsilon = C_2 \left( \sqrt{\frac{\log(1/\delta)}{n}} \right)^{\beta/D} + \epsilon_0$ We have by Assumption 1,

$$\check{r}(x, \epsilon_0) \geq \left( \frac{1}{\hat{C}_\beta} \left( \epsilon_0 + C_2 \left( \sqrt{\frac{\log(1/\delta)}{n}} \right)^{\beta/D} \right) \right)^{1/\beta} - \left( \frac{1}{\check{C}_\beta} \left( C_2 \left( \sqrt{\frac{\log(1/\delta)}{n}} \right)^{\beta/D} \right) \right)^{1/\beta}.$$

Then, there exists constant $C'' > 0$ sufficiently large depending on $f$ such that if

$$\epsilon_0 \geq C'' \left( \left( \sqrt{\frac{\log(1/\delta)}{n}} \right)^{\beta/D} + \frac{\log(1/\delta)\sqrt{\log n}}{\sqrt{k}} \right)$$

then the conditions in Lemma 3 are satisfied for $n$ sufficiently large. Thus, we have for all $x \in \mathcal{X}$ with $f(x) \geq \alpha - C_2 \left( \sqrt{\frac{\log(1/\delta)}{n}} \right)^{\beta/D}$, then $f_k(x) \geq \alpha - \epsilon$. Hence, $\widehat{H}_\alpha(f) \subseteq H_\alpha^{upper}(f)$.

We now do the same in the other direction. Define
$$H_\alpha^{lower}(f) := \{x \in X : f_k(x) \geq \lambda_\alpha + \epsilon\},$$
where $\epsilon$ will be chosen such that $H_\alpha^{lower}(f) \subseteq \widehat{H}_\alpha(f)$. By Lemma 4, it suffices to show that if $f_k(x) \geq \lambda_\alpha + \epsilon$ then $f(x) \geq \lambda_\alpha + C_2 \left( \sqrt{\frac{\log(1/\delta)}{n}} \right)^{\beta/D}$. This direction follows a similar argument as the previous.

Thus, there exists a constant $C_1 > 0$ depending on $f$ such that for $n$ sufficiently large depending on $f$, we have:

$$|\widehat{\lambda}_\alpha - \lambda_\alpha| \leq C_1 \left( \left( \sqrt{\frac{\log(1/\delta)}{n}} \right)^{\beta/D} + \frac{\log(1/\delta)\sqrt{\log n}}{\sqrt{k}} \right),$$

as desired. $\qquad\square$

The next result bounds $\widehat{H}_\alpha(f)$ between two level sets of $f$.

**Lemma 6.** *Let $0 < \delta < 1$. There exists constant $C_1 > 0$ depending on $f$ such that the following holds with probability at least $1 - \delta$ for $n$ sufficiently large depending on $f$. Define*

$$H_\alpha^U(f) := \left\{ x \in \mathcal{X} : f(x) \geq \lambda_\alpha - C_1 \left( \left( \sqrt{\frac{\log(1/\delta)}{n}} \right)^{\beta/D} + \frac{\log(1/\delta)\sqrt{\log n}}{\sqrt{k}} \right) \right\}$$

$$H_\alpha^L(f) := \left\{ x \in \mathcal{X} : f(x) \geq \lambda_\alpha + C_1 \left( \left( \sqrt{\frac{\log(1/\delta)}{n}} \right)^{\beta/D} + \frac{\log(1/\delta)\sqrt{\log n}}{\sqrt{k}} \right) \right\}.$$

*Then,*

$$H_\alpha^L(f) \cap X \subseteq \widehat{H}_\alpha(f) \subseteq H_\alpha^U(f) \cap X.$$

*Proof.* To simplify notation, let us define the following:

$$K(n,k,\delta) := \left(\left(\sqrt{\frac{\log(1/\delta)}{n}}\right)^{\beta/D} + \frac{\log(1/\delta)\sqrt{\log n}}{\sqrt{k}}\right).$$

By Lemma 5, there exists $C_2 > 0$ such that defining

$$\widehat{H_\alpha^U}(f) := \{x \in X : f_k(x) \geq \lambda_\alpha - C_2 \cdot K(n,k,\delta)\}$$
$$\widehat{H_\alpha^L}(f) := \{x \in X : f_k(x) \geq \lambda_\alpha + C_2 \cdot K(n,k,\delta)\},$$

then we have

$$\widehat{H_\alpha^L}(f) \subseteq \widehat{H_\alpha}(f) \subseteq \widehat{H_\alpha^U}(f).$$

It suffices to show that there exists a constant $C_1 > 0$ such that

$$H_\alpha^L(f) \cap X \subseteq \widehat{H_\alpha^L}(f) \text{ and } \widehat{H_\alpha^U}(f) \subseteq H_\alpha^U(f) \cap X.$$

We start by showing $H_\alpha^L(f) \cap X \subseteq \widehat{H^L}_\alpha(f)$. To do this, show that for any $x \in \mathcal{X}$ satisfying $f(x) \geq \lambda_\alpha + C_1 \cdot K(n,k,\delta) + \epsilon$ satisfies $f_k(x) \geq \lambda_\alpha + C_1 \cdot K(n,k,\delta)$, where $\epsilon > 0$ will be chosen later. By a similar argument as in the proof of Lemma 5, we can choose $\epsilon \geq C' \cdot K(n,k,\delta)$ for some constant $C' > 0$ and the desired result holds for $n$ sufficiently large. Similarly, there exists $C'' > 0$ such that $f_k(x) \leq \lambda_\alpha - (C_1 + C'') \cdot K(n,k,\delta)$ implies that $f(x) \leq \lambda_\alpha - C_1 \cdot K(n,k,\delta)$. The result follows by taking $C_2 = C_1 + \max\{C', C''\}$. $\qquad\square$

We are now ready to prove Theorem 5, a more general version of Theorem 1 which makes the dependence on $\delta$ explicit. Note that if $\delta = 1/n$, then $\log(1/\delta) = \log(n)$.

**Theorem 5.** *[Extends Theorem 1] Let $0 < \delta < 1$ and suppose that $f$ is continuous and has compact support $\mathcal{X} \subseteq \mathbb{R}^D$ and satisfies Assumption 1. There exists constants $C_l, C_u, C > 0$ depending on $f$ such that the following holds with probability at least $1 - \delta$. Suppose that $k$ satisfies*

$$C_l \cdot \log(1/\delta)^2 \cdot \log n \leq k \leq C_u \cdot \log(1/\delta)^{2D/(2\beta+D)} \cdot (\log n)^{D(2\beta+D)} \cdot n^{2\beta/(2\beta+D)},$$

*then we have*

$$d_H(H_\alpha(f), \widehat{H_\alpha}(f)) \leq C \cdot \left(\log(1/\delta)^{1/2D} \cdot n^{-1/2D} + \log(1/\delta)^{1/\beta} \cdot \log(n)^{1/2\beta} \cdot k^{-1/2\beta}\right).$$

*Proof of Theorem 5.* Again, to simplify notation, let us define the following:

$$K(n,k,\delta) := \left(\left(\sqrt{\frac{\log(1/\delta)}{n}}\right)^{\beta/D} + \frac{\log(1/\delta)\sqrt{\log n}}{\sqrt{k}}\right).$$

There are two directions to show for the Hausdorff distance result. That (i) $\max_{x \in \widehat{H_\alpha}(f)} d(x, H_\alpha(f))$ is bounded, that is none of the high-density points recovered by Algorithm 1 are far from the true high-density region; and (ii) that $\sup_{x \in H_\alpha(f)} d(x, \widehat{H_\alpha}(f))$ is bounded, that Algorithm 1 recovers a good covering of the entire high-density region.

We first show (i). By Lemma 6, we have that there exists $C_1 > 0$ such that

$$H_\alpha^U(f) := \{x \in \mathcal{X} : f(x) \geq \lambda_\alpha - C_1 K(n,k,\delta)\}$$

contains $\widehat{H_\alpha}(f)$. Thus,

$$\max_{x \in \widehat{H_\alpha}(f)} d(x, H_\alpha(f)) \leq \sup_{x \in H_\alpha^U(f)} d(x, H_\alpha(f)) \leq \left(C_1 \cdot K(n,k,\delta) \cdot \frac{1}{\check{C}_\beta}\right)^{1/\beta},$$

where the second inequality holds by Assumption 1. Now for the other direction, we have by triangle inequality that

$$\sup_{x \in H_\alpha(f)} d(x, \widehat{H_\alpha}(f)) \leq \sup_{x \in H_\alpha(f)} d(x, H_\alpha^L(f)) + \sup_{x \in H_\alpha^L(f)} d(x, \widehat{H_\alpha}(f)).$$

The first term can be bounded by using Assumption 1:

$$\sup_{x \in H_\alpha(f)} d(x, H_\alpha^L(f)) \leq \left( C_1 \cdot K(n, k, \delta) \cdot \frac{1}{\check{C}_\beta} \right)^{1/\beta}.$$

Now for the second term, we see that by Lemma 6, $\widehat{H_\alpha}(f)$ contains all of the sample points of $H_\alpha^L(f)$. Thus, we have

$$\sup_{x \in H_\alpha^L(f)} d(x, \widehat{H_\alpha}(f)) \leq \sup_{x \in H_\alpha^L(f)} d(x, H_\alpha^L(f) \cap X).$$

By Assumption 1, for $r < r_0$, and $x \in H_\alpha^L(f)$ we have $\mathcal{F}(B(x, r)) \geq \rho r^D$, where $\mathcal{F}$ is the distribution corresponding to $f$. Choosing $r \geq \left( \frac{C_{\delta,n}}{\rho} \frac{\sqrt{D \log n}}{n} \right)^{1/D}$ gives us that by Lemma 1 that $\mathcal{F}_n(B(x, r)) > 0$ where $\mathcal{F}_n$ is the distribution of $X$ and thus, we have

$$\sup_{x \in H_\alpha^L(f)} d(x, H_\alpha^L(f) \cap X) \leq \left( \frac{C_{\delta,n}}{\rho} \frac{\sqrt{D \log n}}{n} \right)^{1/D},$$

which is dominated by the error contributed by the other error and the result follows. $\qquad \square$

## C  Supporting results for Theorem 2 Proof

In this section, we note that we will reuse some notation from the last section for the manifold case.

**Lemma 7** (Manifold version of uniform convergence of empirical Euclidean balls (Lemma 7 of [46]))**.** *Let $\mathcal{F}$ be the true distribution and $\mathcal{F}_n$ be the empirical distribution w.r.t. sample $X$. Let $\mathcal{N}$ be a minimal fixed set such that each point in $M$ is at most distance $1/n$ from some point in $\mathcal{N}$. There exists a universal constant $C_0$ such that the following holds with probability at least $1 - \delta$. For all $x \in X \cup \mathcal{N}$,*

$$\mathcal{F}(B) \geq C_{\delta,n} \frac{\sqrt{d \log n}}{n} \Rightarrow \mathcal{F}_n(B) > 0$$

$$\mathcal{F}(B) \geq \frac{k}{n} + C_{\delta,n} \frac{\sqrt{k}}{n} \Rightarrow \mathcal{F}_n(B) \geq \frac{k}{n}$$

$$\mathcal{F}(B) \leq \frac{k}{n} - C_{\delta,n} \frac{\sqrt{k}}{n} \Rightarrow \mathcal{F}_n(B) < \frac{k}{n},$$

*where $C_{\delta,n} = C_0 \log(2/\delta) \sqrt{d \log n}$, $\mathcal{F}_n$ is the empirical distribution, and $k \geq C_{\delta,n}$.*

**Definition 5** (k-NN Density Estimator on Manifold)**.**

$$f_k(x) := \frac{k}{n \cdot v_d \cdot r_k(x)^d}.$$

**Lemma 8** (Manifold version of $f_k$ upper bound [42])**.** *Define the following which charaterizes how much the density increases locally in $M$:*

$$\hat{r}(\epsilon, x) := \sup \left\{ r : \sup_{x' \in B(x,r) \cap M} f(x') - f(x) \leq \epsilon \right\}.$$

*Fix $\lambda_0 > 0$ and $\delta > 0$ and suppose that $k \geq C_{\delta,n}^2$. Then there exists constant $C_1 \equiv C_1(\lambda_0, d, \tau)$ such that if*

$$k \leq C_1 \cdot C_{\delta,n}^{2d/(2+d)} \cdot n^{2/(2+d)},$$

*then the following holds with probability at least $1 - \delta$ uniformly in $\epsilon > 0$ and $x \in X$ with $f(x) + \epsilon \geq \lambda_0$:*

$$f_k(x) < \left( 1 + 3 \cdot \frac{C_{\delta,n}}{\sqrt{k}} \right) \cdot (f(x) + \epsilon),$$

*provided $k$ satisfies $v_d \cdot \hat{r}(\epsilon, x)^d \cdot (f(x) + \epsilon) \geq \frac{k}{n} - C_{\delta,n} \frac{\sqrt{k}}{n}$.*

**Lemma 9** (Manifold version of $f_k$ lower bound [42])**.** *Define the following which charaterizes how much the density decreases locally in $M$:*

$$\check{r}(\epsilon, x) := \sup \left\{ r : \sup_{x' \in B(x,r) \cap M} f(x) - f(x') \le \epsilon \right\}.$$

*Fix $\lambda_0 > 0$ and $0 < \delta < 1$ and suppose $k \ge C_{\delta,n}$. Then there exists constant $C_2 \equiv C_2(\lambda_0, d, \tau)$ such that if*

$$k \le C_2 \cdot C_{\delta,n}^{2d/(4+d)} \cdot n^{4/(4+d)},$$

*then with probability at least $1 - \delta$, the following holds uniformly for all $\epsilon > 0$ and $x \in X$ with $f(x) - \epsilon \ge \lambda_0$:*

$$f_k(x) \ge \left( 1 - 3 \cdot \frac{C_{\delta,n}}{\sqrt{k}} \right) \cdot (f(x) - \epsilon),$$

*provided $k$ satisfies $v_d \cdot \check{r}(\epsilon, x)^d \cdot (f(x) - \epsilon) \ge \frac{4}{3} \left( \frac{k}{n} + C_{\delta,n} \frac{\sqrt{k}}{n} \right)$.*

## D   Proof of Theorem 2

The proof essentially follows the same structure as the full-dimensional case, with the primary difference in the density estimation bounds.

**Lemma 10** (Manifold Version of Lemma 4)**.** *There exists constants $C_1, r_1 > 0$ depending on $f$ such that the following holds for all $0 < \epsilon < r_1$ such that*

$$0 < \lambda_\alpha - \lambda_{\alpha-\epsilon} \le C_1 \epsilon^{\beta/d} \text{ and } 0 < \lambda_{\alpha+\epsilon} - \lambda_\alpha \le C_1 \epsilon^{\beta/d}.$$

*Proof.* The proof follows the same structure as the proof of Lemma 4, with the difference being the change in dimension, and is omitted here. □

**Lemma 11** (Manifold Version of Lemma 5)**.** *Let $0 < \delta < 1$. Let $\widehat{\varepsilon}$ be the $\varepsilon$ setting chosen by Algorithm 1 after the binary search procedure. Define*

$$\widehat{\lambda_\alpha} := \frac{k}{v_D \cdot n \cdot \widehat{\varepsilon}^d}.$$

*Then, with probability at least $1 - \delta$, we have there exist constant $C_1 > 0$ depending on $f$ and $M$ such that for $n$ sufficiently large depending on $f$ and $M$, we have*

$$|\widehat{\lambda}_\alpha - \lambda_\alpha| \le C_1 \left( \left( \sqrt{\frac{\log(1/\delta)}{n}} \right)^{\beta/d} + \frac{\log(1/\delta)\sqrt{\log n}}{\sqrt{k}} \right).$$

*Proof.* The proof is essentially the same as that of Lemma 5. The only difference is that instead of applying the full-dimensional versions of the uniform $k$-NN density estimate bounds (Lemma 2 and 3), we instead apply the manifold analogues (Lemma 8 and 9). Asides from constant factors, the major difference is in the allowable range for $k$. In the full-dimensional case, we only need $k \lesssim n^{2\beta/(2\beta+d)}$ for the density estimation bounds to hold. However, here we require $k \lesssim \min\{n^{2/(2+d)}, n^{2\beta/(2\beta+d)}\} = n^{2\max\{1,\beta\}/(2\beta'+d)}$. □

**Lemma 12** (Manifold Version of Lemma 5)**.** *Let $0 < \delta < 1$. There exists constant $C_1 > 0$ depending on $f$ and $M$ such that the following holds with probability at least $1 - \delta$ for $n$ sufficiently large depending on $f$ and $M$. Define*

$$H_\alpha^U(f) := \left\{ x \in \mathcal{X} : f(x) \ge \lambda_\alpha - C_1 \left( \left( \sqrt{\frac{\log(1/\delta)}{n}} \right)^{\beta/d} + \frac{\log(1/\delta)\sqrt{\log n}}{\sqrt{k}} \right) \right\}$$

$$H_\alpha^L(f) := \left\{ x \in \mathcal{X} : f(x) \ge \lambda_\alpha + C_1 \left( \left( \sqrt{\frac{\log(1/\delta)}{n}} \right)^{\beta/d} + \frac{\log(1/\delta)\sqrt{\log n}}{\sqrt{k}} \right) \right\}.$$

*Then,*

$$H_\alpha^L(f) \cap X \subseteq \widehat{H_\alpha}(f) \subseteq H_\alpha^U(f) \cap X.$$

*Proof.* Same comment as the proof for Lemma 11. □

**Theorem 6.** *[Extends Theorem 2] Let $0 < \delta < 1$. Suppose that density function $f$ is continuous and supported on $M$ and Assumptions 1 and 2 hold. Suppose also that there exists $\lambda_0 > 0$ such that $f(x) \geq \lambda_0$ for all $x \in M$. Then, there exists constants $C_l, C_u, C > 0$ depending on $f$ such that the following holds with probability at least $1 - \delta$. Suppose that $k$ satisfies,*

$$C_l \cdot \log(1/\delta)^2 \cdot \log n \leq k \leq C_u \cdot \log(1/\delta)^{2d/(2\beta'+d)} \cdot (\log n)^{d(2\beta'+d)} \cdot n^{2\beta'/(2\beta'+d)}$$

*where $\beta' := \max\{1, \beta\}$. Then we have*

$$d_H(H_\alpha(f), \widehat{H_\alpha}(f)) \leq C \cdot \left( \log(1/\delta)^{1/2d} \cdot n^{-1/2d} + \log(1/\delta)^{1/\beta} \cdot \log(n)^{1/2\beta} \cdot k^{-1/2\beta} \right).$$

*Proof of Theorem 6.* Proof is the same as the full-dimensional case given the contributed Lemmas of this section and is omitted here. □

# E    Supporting Results for Theorem 3 Proof

Next, we need the following on the volume of the intersection of the Euclidean ball and $M$; this is required to get a handle on the true mass of the ball under $\mathcal{F}_M$ in later arguments. The upper and lower bounds follow from [57] and Lemma 5.3 of [44]. The proof can be found e.g. in [42].

**Lemma 13** (Ball Volume). *If $0 < r < \min\{\tau/4d, 1/\tau\}$, and $x \in M$ then*

$$v_d r^d (1 - \tau^2 r^2) \leq vol_d(B(x, r) \cap M) \leq v_d r^d (1 + 4dr/\tau),$$

*where $v_d$ is the volume of a unit ball in $\mathbb{R}^d$ and $vol_d$ is the volume w.r.t. the uniform measure on $M$.*

The next is a bound uniform convergence of balls:

**Lemma 14** (Lemma 3 of [58]). *Let $\mathcal{B}$ be the set of all balls in $\mathbb{R}^D$, $\mathcal{F}$ is some distribution and $\mathcal{F}_n$ is an empirical distribution. With probability at least $1 - \delta$, the following holds uniformly for every $B \in \mathcal{B}$ and $\gamma \geq 0$:*

$$\mathcal{F}(B) \geq \gamma \Rightarrow \mathcal{F}_n(B) \geq \gamma - \beta_n \sqrt{\gamma} - \beta_n^2,$$
$$\mathcal{F}(B) \leq \gamma \Rightarrow \mathcal{F}_n(B) \leq \gamma + \beta_n \sqrt{\gamma} + \beta_n^2,$$

*where $\beta_n = 8d \log(1/\delta)\sqrt{\log n/n}$.*

# F    Proof of Theorem 3

The first result says that within the manifold, the vast majority of the probability mass is attributed to the manifold distributions.

**Lemma 15.** *There exists $C_1, r_1 > 0$ depending on $\mathcal{F}_M, \mathcal{F}_E, M$ such that the following holds uniformly over $x \in M$ and $0 < r < r_1$.*

$$\frac{\mathcal{F}_E(B(x,r))}{\mathcal{F}_M(B(x,r))} \leq C_1 \cdot r^{D-d}.$$

*Proof.* Let $x \in M$ and $r > 0$. We have

$$\mathcal{F}_M(B(x,r)) \geq \lambda_0 \cdot vol_d(B(x,r) \cap M) \geq v_d r^d (1 - \tau^2 r^2) \cdot \lambda_0,$$

where the second inequality holds by Lemma 13 for $r$ sufficiently small. On the other hand, we have

$$\mathcal{F}_E(B(x,r)) \leq ||f_E||_\infty v_D r^D.$$

Thus, we have there exists $C_1 > 0$ depending on $f_M$, $M$, and $f_E$ such that

$$\frac{\mathcal{F}_E(B(x,r))}{\mathcal{F}_M(B(x,r))} \leq C_1 \cdot r^{D-d},$$

as desired. □

We next show that points far away from $H_{\widetilde{\alpha}}(f_M)$ do not get selected as high-density points by Algorithm 1.

**Lemma 16.** *There exists $\omega_0 > 0$ such that for any $0 < \omega < \omega_0$ and $n$ sufficiently large depending on $\mathcal{F}_M, \mathcal{F}_E, M$ and $\omega$, with probability at least $1 - \delta$, Algorithm 1 will not select any points outside of $H_{\widetilde{\alpha}-\omega}(f_M)$.*

*Proof.* By Assumption 1, we can choose $\omega$ sufficiently small so that for the density $f_M$, $|\lambda_{\widetilde{\alpha}-\omega} - \lambda_{\widetilde{\alpha}}| \le \check{C}_\beta \cdot (r_c/3)^\beta$. Then, at the $(\widetilde{\alpha}-\omega)$-density level, we will be within the area where the regularity assumptions hold.

Next, by Hoeffding's inequality, we have that there exist constant $C' > 0$ such that for $\bar{\alpha} > 0$:

$$\mathbb{P}\left(1 - \bar{\alpha} - C'\sqrt{\frac{\log(1/\delta)}{n}} \le \frac{|H_{\frac{\widetilde{\alpha}-\eta}{1-\eta}}(f_M) \cap X|}{n} \le 1 - \bar{\alpha} + C'\sqrt{\frac{\log(1/\delta)}{n}}\right) \ge 1 - \delta/3.$$

Choosing $\bar{\alpha} = \alpha - C'\sqrt{\frac{\log(1/\delta)}{n}}$, then it follows that with probability at least $1 - \delta/3$,

$$H_0 := H_{\widetilde{\alpha}-C'\sqrt{\log(1/\delta)}/(\sqrt{n}\cdot(1-\eta))}(f_M)$$

satisfies $|H_0 \cap X| > (1 - \alpha) \cdot n$. Next let

$$H_\omega := H_{\widetilde{\alpha}-\omega}(f_M).$$

Let $r$ be the value of $\varepsilon$ used by Algorithm 1. Now, it suffices to show that for $n$ sufficiently large depending on $f_M$:

$$\max_{x \in X \backslash H_\omega} \mathcal{F}_n(B(x,r)) < \min_{x \in H_0} \mathcal{F}_n(B(x,r)),$$

where $\mathcal{F}_n$ is the empirical distribution. This is because Algorithm 1 filters out sample points whose $\varepsilon$-ball has less than $k$ sample points for its final $\varepsilon$ value, which is the value which allows it to filter $\alpha$-fraction of the points.

By Lemma 15, it suffices to show that

$$\max_{x \in X \backslash H_\omega} \mathcal{F}_{M,n}(B(x,r))\left(1 + C_1 r^{D-d}\right) < \min_{x \in H_0} \mathcal{F}_{M,n}(B(x,r))\left(1 - C_1 r^{D-d}\right),$$

where $\mathcal{F}_{M,n}(A)$ denote the fraction of samples drawn from $\mathcal{F}_M$ which lie in $A$ w.r.t. our entire sample $X$.

Then, by Lemma 14, it will be enough to show that

$$\max_{x \in X \backslash H_\omega}\left(\mathcal{F}_M(B(x,r)) + \beta_n\sqrt{\mathcal{F}_M(B(x,r))} + \beta_n^2\right)\left(1 + C_1 r^{D-d}\right)$$
$$< \min_{x \in H_0}\left(\mathcal{F}_M(B(x,r)) - \beta_n\sqrt{\mathcal{F}_M(B(x,r))} - \beta_n^2\right)\left(1 - C_1 r^{D-d}\right),$$

where $\beta_n = 8d\log(1/\delta)\sqrt{\log n/n}$.

To bound the LHS, we have by Lemma 13

$$\max_{x \in X \backslash H_\omega}\left(\mathcal{F}_M(B(x,r)) + \beta_n \cdot \sqrt{\mathcal{F}_M(B(x,r))} + \beta_n^2\right)\left(1 + C_1 r^{D-d}\right)$$
$$\le \max_{x \in X \backslash H_\omega}\left\{\left(\inf_{x' \in B(x,r)} f_M(x')\right) \cdot \left(1 + \beta_n/\sqrt{||f_M||_\infty} + \beta_n^2/||f_M||_\infty\right)\left(1 + C_1 r^{D-d}\right)(1 + 4dr/\tau)\right\}$$
$$\le \max_{x \in X \backslash H_\omega}\left\{\left(\inf_{x' \in B(x,r)} f_M(x')\right)(1 + C_2\beta_n + C_3 r)\right\}$$
$$\le \left(\lambda_{\widetilde{\alpha}-\omega} + \iota(f_M, r)\right)(1 + C_2\beta_n + C_3 r),$$

for some $C_2, C_3 > 0$ and $\iota$ is the modulus of continuity, that is $\iota(f_M, r) := \sup_{x,x' \in M:|x-x'|\le r} |f_M(x) - f_M(x')|$ (i.e. $f_M$ is uniformly continuous since it is continuous over a compact support, so $\iota(f_M, r) \to 0$ as $r \to 0$).

Similarly, for the RHS, we can show for some constants $C_4, C_5$ that

$$\min_{x \in H_0} \left( \mathcal{F}_M(B(x,r)) - \beta_n \sqrt{\mathcal{F}_M(B(x,r))} - \beta_n^2 \right) \left( 1 - C_1 r^{D-d} \right)$$
$$\geq \left( \lambda_{\widetilde{\alpha} - C' \sqrt{\log(1/\delta)}/(\sqrt{n}(1-\eta))} - \iota(f_M, r) \right) \left( 1 - C_4 \beta_n - C_5 r \right).$$

The result follows since $r \to 0$ as $n \to \infty$ (since by Lemma 1, $r$ is a $k$-NN radius so $r \lesssim (k/n)^{1/D} \to 0$ given the conditions on $k$ of Theorem 3) and the fact that $\lambda_{\widetilde{\alpha} - \omega} < \lambda_{\widetilde{\alpha} - C' \sqrt{\log(1/\delta)}/\sqrt{n}}$ for $n$ sufficiently large. As desired. $\qquad\square$

**Lemma 17** (Bounding density estimators w.r.t to entire sample vs w.r.t. samples on manifold). *For $x \in \mathbb{R}^D$, define the following:*

$$r_k(x) := \inf\{\epsilon > 0 : |B(x,\epsilon) \cap X| \geq k\}$$
$$\widetilde{r}_k(x) := \inf\{\epsilon > 0 : |B(x,\epsilon) \cap X \cap M| \geq k\}$$

*where the former is simply the $k$-NN radius we've been using thus far and the latter is the $k$-NN radius if we were to restrict the samples to only those that came from $M$. Then likewise, define the analogous density estimators:*

$$f_k(x) := \frac{k}{n \cdot v_d \cdot r_k(x)^d} \quad \text{and} \quad \widetilde{f}_k(x) := \frac{k}{n \cdot v_d \cdot \widetilde{r}_k(x)^d},$$

*where again, the former is the usual $k$-NN density estimator on manifolds. Then, there exists $C_1$ such that the following holds with high probability.*

$$\sup_{x \in M} |f_k(x) - \widetilde{f}_k(x)| \leq C_1 \cdot (k/n)^{D/d-1}.$$

*Proof.* By Lemma 14, there exists $C_2 > 0$ depending on $\mathcal{F}$ and $\mathcal{F}_M$ such that

$$\mathcal{F}_n(B(x, r_k(x))) = k \Rightarrow |\mathcal{F}(B(x, r_k(x))) - k| \leq C_2 \beta_n,$$

and

$$\mathcal{F}_{M,n}(B(x, \widetilde{r}_k(x))) = k \Rightarrow |\mathcal{F}_M(B(x, \widetilde{r}_k(x))) - k| \leq C_2 \beta_n,$$

where $\mathcal{F}_{M,n}$ is the empirical distribution w.r.t. $X \cap M$. Next, by Lemma 15, we have for some constant $C_3 > 0$:

$$\mathcal{F}(B(x, \widetilde{r}_k(x))) \leq \mathcal{F}_M(B(x, \widetilde{r}_k(x)))(1 + C_3 \widetilde{r}_k(x)^{D-d})$$
$$\leq (k + C_2 \beta_n)(1 + C_3 \cdot \widetilde{r}_k(x)^{D-d})$$
$$\leq (k + C_2 \beta_n)(1 + C_4 \cdot (k/n)^{D/d-1})$$
$$\leq k \cdot (1 + C_5 (k/n)^{D/d-1}),$$

where the second last inequality holds for some constant $C_4 > 0$ by Lemma 7 and $C_5 > 0$ is some constant depending on $\mathcal{F}$ and $\mathcal{F}_M$ and $M$. Then it follows that for some constant $C_6 > 0$, we have

$$\frac{\mathcal{F}(B(x, \widetilde{r}_k(x)))}{\mathcal{F}(B(x, r_k(x)))} \leq 1 + C_6 (k/n)^{D/d-1}.$$

In the other direction, we trivially have $\widetilde{r}_k(x) \geq r_k(x)$, so

$$1 \leq \frac{\mathcal{F}(B(x, \widetilde{r}_k(x)))}{\mathcal{F}(B(x, r_k(x)))} \leq 1 + C_6 (k/n)^{D/d-1}.$$

The result follows. $\qquad\square$

**Theorem 7.** *[Extends Theorem 3] Let $0 < \eta < \alpha < 1$ and $0 < \delta < 1$. Suppose that distribution $\mathcal{F}$ is a weighted mixture $(1 - \eta) \cdot \mathcal{F}_M + \eta \cdot \mathcal{F}_E$ where $\mathcal{F}_M$ is a distribution with continous density $f_M$ supported on a $d$-dimensional manifold $M$ satisfying Assumption 2 and $\mathcal{F}_E$ is a (noise) distribution with continuous density $f_E$ with compact support over $\mathbb{R}^D$ with $d < D$. Suppose also that there exists $\lambda_0 > 0$ such that $f_M(x) \geq \lambda_0$ for all $x \in M$ and $H_{\widetilde{\alpha}}(f_M)$ (where $\widetilde{\alpha} := \frac{\alpha - \eta}{1 - \eta}$) satisfies Assumption 1 for density $f_M$. Let $\widehat{H}_\alpha$ be the output of Algorithm 1 on a sample $X$ of size $n$ drawn i.i.d. from $\mathcal{F}$.*

*Then, there exists constants $C_l, C_u, C > 0$ depending on $f_M$, $f_E$, $\eta$, $M$ such that the following holds with probability at least $1 - \delta$. Suppose that $k$ satisfies*

$$C_l \cdot \log(1/\delta)^2 \cdot \log n \leq k \leq C_u \cdot \log(1/\delta)^{2d/(2\beta'+d)} \cdot (\log n)^{d(2\beta'+d)} \cdot n^{2\beta'/(2\beta'+d)}$$

*where $\beta' := \max\{1, \beta\}$. Then we have*

$$d_H(H_{\widetilde{\alpha}}(f_M), \widehat{H_\alpha}) \leq C \cdot \left( \log(1/\delta)^{1/2d} \cdot n^{-1/2d} + \log(1/\delta)^{1/\beta} \cdot \log(n)^{1/2\beta} \cdot k^{-1/2\beta} \right).$$

*Proof of Theorem 7.* The proof follows in a similar way as that of Theorem 6, except with the complexity of having added full-dimensional noise. We will only highlight the difference and provide a sketch of the proof here.

Lemma 16 and 17 give us a handle on the additional complexity when having separate noise distribution, compared to the earlier manifold setting of Theorem 2.

Lemma 16 guarantees that the points in $\widehat{H_\alpha}$ lie in the inside of $M$ with margin. In particular, that means the noise points are filtered out by the algorithm and thus, we are reduced to reasoning about the $\widetilde{\alpha}$-high-density-set of $f_M$.

Then, Lemma 17 ensures that the $k$-NN density estimator used for our analysis for the entire sample $X$ is actually quite close to the $k$-NN density estimator with respect to $M \cap X$ within $M$. In other words, we can use the $k$-NN density estimator to estimate $f_M$ without knowing which samples of $X$ were in $M$. Lemma 17 shows that the additional error in density estimation we obtain is $\approx (k/n)^{D/d-1} \lesssim (k/n)^{1/d} \lesssim (\log n)/\sqrt{k}$, where the first inequality holds since $D > d$ and the latter holds from the conditions on $k$. It turns out that this error term can be absorbed as a constant in the previous result of Theorem 6. $\qquad\square$

## G   Proof of Theorem 4

*Proof of Theorem 4.* For the first inequality, we have

$$\xi(h, x) \geq \frac{d(x, M_{\widetilde{h}(x)}) - \epsilon_n}{d(x, M_{h(x)}) + \epsilon_n} = \frac{d(x, M_{\widetilde{h}(x)})}{d(x, M_{h(x)})} - \frac{\epsilon_n}{d(x, M_{h(x)}) + \epsilon_n} \cdot \left( \frac{d(x, M_{\widetilde{h}(x)})}{d(x, M_{h(x)})} + 1 \right),$$

where the first inequality holds by Theorem 3. This, along with the condition on $\gamma$ and $\varepsilon(h, x)$ fromo the theorem statement, implies that

$$\frac{d(x, M_{\widetilde{h}(x)})}{d(x, M_{h(x)})} < 1 - \gamma,$$

which implies that $h(x) \neq h^*(x)$. For the second inequality, we have

$$\frac{1}{\xi(h, x)} \geq \frac{d(x, M_{h(x)}) - \epsilon_n}{d(x, M_{\widetilde{h}(x)}) + \epsilon_n} = \frac{d(x, M_{h(x)})}{d(x, M_{\widetilde{h}(x)})} - \frac{\epsilon_n}{d(x, M_{\widetilde{h}(x)}) + \epsilon_n} \cdot \left( \frac{d(x, M_{h(x)})}{d(x, M_{\widetilde{h}(x)})} + 1 \right),$$

where the first inequality holds by Theorem 3. Thus, if the condition of the theorem statement holds, then

$$\frac{d(x, M_{h(x)})}{d(x, M_{\widetilde{h}(x)})} < 1 - \gamma \Rightarrow \frac{d(x, M_{h(x)})}{d(x, M_c)} < 1 - \gamma$$

for all $c \neq h(x)$, which implies that $h(x) = h^*(x)$ $\qquad\square$

# H Additional UCI Experiments

## H.1 When to trust: Precision for correct predictions by percentile

Figure 4: UCI data sets and precision on correctness

## H.2 When to not trust: Precision for misclassification predictions by percentile

Figure 5: UCI data sets and precision on incorrectness

## H.3 High dimensional Datasets

(a) MNIST

(b) MNIST

(c) SVHN

(d) SVHN

(e) CIFAR-10

(f) CIFAR-10

(g) CIFAR-100

(h) CIFAR-100

Figure 6: Trust score results using convolutional neural networks on MNIST, SVHN, CIFAR-10, and CIFAR-100 datasets. Left column is detecting trustworthy; right column is detecting suspicious.