[Reviews · NeurIPS 2018]

Reviewer 1



This paper proposes a "trust" score that is supposed to reliably identify whether a prediction is correct or not. The main intuition is that the prediction is more reliable if the instance is closer to the predicted class than other classes. By defining alpha-high-density-set, this paper is able to provide theoretical guarantees of the proposed algorithm, which is the main strength of this paper. This paper proceeds to evaluate the "trust" score using a variety of datasets. One cool experiment is to show that the "trust" score estimated from deeper layers than lower layers. My major concern about this paper lies in the motivation behind the proposed score. Trust is certainly an important problem for our research community to work on. But I find this paper abuses this term, "trust". In general, trust should involve some notion of humans/users/agents. One definition in Lee and See, 2004 [1] of trust is "the attitude that an agent will help achieve an individual’s goals in a situation characterized by uncertainty and vulnerability". It is concerning that the "trust" score in this paper does not involve any consideration of the end user. For instance, the score will fall in any range and it is unclear that users or experts can determine whether they should trust the classifier based on this score. A percentile adaptation could help alleviate this issue, which is how this paper conducts the evaluation. However, although the percentile adaptation seems to make the proposed score more interpretable, it carries very different information from calibration: the proposed score only guarantees that instances with high "trust" scores should be more likely to be correct, but the 90 percentile in "trust" score does not necessarily lead to 90% accuracy. This further shows that it can be challenging for any agent to make sense of the proposed "trust" score, even though the precision-percentile curves look nice. Another difference from calibration that this paper emphasizes is that the ordering preserving property, but this property seems to be a limitation of the current calibration approach. The goal of calibration is that instances with a certain confidence score lead to the same accuracy. So a calibrated classifier may not optimize this precision-percentile curve. Again, I do not find the notion of "trust" score in this paper well motivated. This paper would have been stronger if it can articulate the notion of "trust" and its motivation and difference with confidence with the subject of trusting in mind. Some relatively minor concerns about the experiment that requires clarification/explanation: 1) How is the percentile computed? Based on the description, it seems that the percentile is determined based on the test set, which seems rather strange. It would be natural to determine the percentile based on the training set. 2) The interpretation of percentile seems to be different from the normal interpretation. Does the highest trust score fall in the 100 percentile or the 0 percentile? 3) What are the chosen alphas? It seems that how well this metric behaves is highly dependent on the geometry space. One can also build a modified nearest-neighbor classifier based on the proposed score. How is the performance of the proposed score in the experiments related to the performance of that modified nearest-neighbor classifier? 4) Interpretable machine learning seems to be an area that is clearly related, but not really cited in the references. For instance, Why should I trust you?: Explaining the Predictions of Any Classifier by Ribeiro, Singh, and Guestrin; Examples are not Enough, Learn to Criticize! Criticism for Interpretability by Kim, Khanna, and Koyejo. Also, references are not sorted alphabetically. [1] John D. Lee and Katrina A. See, "Trust in Automation: Designing for Appropriate Reliance", in Human Factors. I have read the rebuttal. I think that this paper is onto something interesting, but it fails to articular what the proposed ranking is for and how it is superior/complementary to calibration from a human perspective. I would caution against calling this trust.

Reviewer 2



The paper presents a new score to estimate the accuracy of any classifier. Knowing when to trust or not to trust a model output is important for many application of machine learning models, thus this task is well motivated. The paper is well structured and well written. The paper first introduces the goal, covers related work, then introduce its algorithm for computing the new score: the trust score. The trust score is defined as the ratio between the distance from the testing sample to the alpha-high-density-set of the nearest class different from the predicted, and the distance from the test sample to the alpha-high-density-set of the class predicted by the classifier. Algorithm 1 covers how to estimate alpha-high-density set, and algorithm 2 covers how to compute the trust score. The section 4 gives theoretical guarantees of both algorithms. To be honest, I was not able to follow all the proofs in the appendix, making it hard for me to properly evaluate this paper. The section 5 gives empirical performance of the trust scores across different datasets, different classifiers, and different representation of the dataset. The evaluation is easy to understand and was persuasive. All the graph in Figure 1 and Figure 2 shows that the proposed “trust score” outperforms either the model performance or 1-NN ratio. Questions to the authors: - How is B(x, r) defined in the algorithm 1? - Does this method apply as is to a binary classification case? If it works better than the model’s confidence score, this implies that this can improve model’s accuracy at varying threshold. - How sensitive is the method to hyper parameters, especially to k? - Is the model performances on par with the state-of-the-art for these tasks? (Not important, but would be good to include) - when the axis are the same, ‘when to predict’ and ‘when not to predict’ is simply inverse of each other? Minor comments: - Some papers, like 12, is cited with ArXiv instead of its official venue (ICLR). Please fix accordingly.

Reviewer 3



This is a unique paper that, from a safety perspective, examines the question of whether a classifier output should be trusted. In particular, it questions the use of confidence scores for that purpose and proposes an alternative. This general direction falls very much under the "safe fail" paradigm for machine learning espoused in https://www.liebertpub.com/doi/abs/10.1089/big.2016.0051, and is clearly important as machine learning algorithms are increasingly used in high-stakes decision making. The fact that it uses topological data analysis and level set estimation towards this end is quite appropriate and brings forth some nice theory. I like that the authors have considered all three of the settings they have in developing their estimation rate results because they build upon each other to get to the most realistic third setting. The analysis does appear to be correct, and I agree with the "remarkable"-ness of the final results. The empirical work seems to be done in a reasonable manner and is sufficient for this type of paper. I like that the algorithms themselves are quite simple and straightforward. Overall, well done piece of work.